# Understanding of Active Sites and Interconversion of Pd and PdO during CH_4_ Oxidation

**DOI:** 10.3390/molecules28041957

**Published:** 2023-02-18

**Authors:** Dong Gun Oh, Hristiyan A. Aleksandrov, Haneul Kim, Iskra Z. Koleva, Konstantin Khivantsev, Georgi N. Vayssilov, Ja Hun Kwak

**Affiliations:** 1School of Energy and Chemical Engineering, Ulsan National Institute of Science and Technology (UNIST), 50 UNIST-gil, Ulsan 44919, Republic of Korea; 2Faculty of Chemistry and Pharmacy, University of Sofia, 1126 Sofia, Bulgaria; 3Institute for Integrated Catalysis, Pacific Northwest National Laboratory, Richland, WA 99352, USA

**Keywords:** active site, CH_4_ oxidation, DRIFTS, PdO_x_

## Abstract

Pd-based catalysts are widely used in the oxidation of CH_4_ and have a significant impact on global warming. However, understanding their active sites remains controversial, because interconversion between Pd and PdO occurs consecutively during the reaction. Understanding the intrinsic active sites under reaction conditions is critical for developing highly active and selective catalysts. In this study, we demonstrated that partially oxidized palladium (PdO_x_) on the surface plays an important role for CH_4_ oxidation. Regardless of whether the initial state of Pd corresponds to oxides or metallic clusters, the topmost surface is PdO_x_, which is formed during CH4 oxidation. A quantitative analysis using CO titration, diffuse reflectance infrared Fourier-transform spectroscopy, X-ray diffraction, and scanning transmission electron microscopy demonstrated that a surface PdO layer was formed on top of the metallic Pd clusters during the CH_4_ oxidation reaction. Furthermore, the time-on-stream test of CH_4_ oxidation revealed that the presence of the PdO layer on top of the metallic Pd clusters improves the catalytic activity. Our periodic density functional theory (DFT) calculations with a PdO_x_ slab and nanoparticle models aided the elucidation of the structure of the experimental PdO particles, as well as the experimental C-O bands. The DFT results also revealed the formation of a PdO layer on the metallic Pd clusters. This study helps achieve a fundamental understanding of the active sites of Pd and PdO for CH_4_ oxidation and provides insights into the development of active and durable Pd-based catalysts through molecular-level design.

## 1. Introduction

Natural gas, mainly consisting of methane, has received considerable attention as an alternative fuel for automobiles owing to its high energy efficiency. Furthermore, it is environmentally friendly, because it generates fewer undesirable by-products per unit energy when compared with other fossil fuels, such as gasoline and diesel. However, unburned CH_4_ contributes significantly to global warming due to its ~20 times higher greenhouse potential than that of carbon dioxide. Therefore, significant research efforts have been dedicated to eliminating CH_4_ from exhaust gas.

Pd-supported catalysts are the most widely used catalysts for methane oxidation because of their high catalytic activity [1,2,3,4,5,6,7,8]. Despite significant research on catalytic active sites over the past decade, the understanding of the nature of the active sites on Pd-supported catalysts remains limited. Müller et al. proved the role of lattice oxygen on PdO nanoparticles (NPs) using labeled ^18^O, suggesting that CH_4_ oxidation follows the Mars-van Krevelen mechanism [9]. Iglesia et al. reported that the CH_4_ oxidation reaction occurs via the redox cycle of the PdO surface [10,11]. Jang et al. demonstrated that the PdO_x_ species generated under the reaction conditions exhibited a linear correlation with CH_4_ oxidation activity on Pd and Pt−Pd bimetallic catalysts [12]. On the contrary, Lyubovsky et al. suggested that metallic Pd is more active toward methane oxidation than PdO at high temperatures [13]. Xu et al. also proposed that CH_4_ oxidation occurs on the metallic Pd surface via the Langmuir–Hinshelwood mechanism, and the stability of metallic Pd is important for CH_4_ oxidation activity at low temperatures [14]. Hellman et al. reported that CH_4_ easily dissociates from the partially reduced PdO(101) or metallic Pd rather than from PdO(100), using density function theory (DFT) calculations [15]. Furthermore, Zorn et al. reported that fully oxidized PdO did not participate in the reaction [16], and Mehar et al. reported that an intrinsic active site for Pd-based catalysts is not single-layer PdO(101) but multilayer PdO(101) [17].

Despite numerous studies on the active sites for CH_4_ oxidation, the understanding of their nature remains limited. In this study, we investigated the behavior of Pd and PdO under the reaction conditions to establish the active sites for CH_4_ oxidation. An investigation using CO titration, diffuse reflectance infrared Fourier-transform spectroscopy (DRIFTS), and in situ X-ray diffraction (XRD) revealed that the initially formed PdO does not change during the CH_4_ oxidation reaction. However, when it started from the metallic Pd cluster, a surface PdO layer gradually formed on the metallic Pd core during CH4 oxidation. Notably, regardless of whether the initial state of Pd corresponds to oxides or metallic clusters, the topmost surface is PdO_x_, suggesting that it is most probably the active site, and it is crucial for the activity and durability of CH_4_ oxidation. To understand the structure of the experimental PdO particles, we identified the most stable positions for the O species on the surface and subsurface regions of the Pd(111) slab models at different coverages. In addition, we also interpretated the DRIFTS results of CO on PdO by employing Pd_79_ nanoparticle models in our periodic density functional theory (DFT) calculations. This study helps to achieve a fundamental understanding of the active sites for CH_4_ oxidation and provides insights into the development of active and durable Pd-based catalysts through molecular-level designs.

## 2. Results and Discussion

### 2.1. Characterization of Pd/Al_2_O_3_ Catalysts

To understand the behavior of Pd and PdO under reaction conditions for CH_4_ oxidation, we prepared 5.5 wt% Pd/Al_2_O_3_ via hydrothermal aging (HTA) at 850 °C and characterized them using XRD and STEM after oxidative and reductive pretreatments at 500 °C, as shown in Figure 1 and Figure 2. The oxidative pretreatment is denoted as “O_x_”, and the reductive pretreatment is denoted as “Red”. The XRD patterns display the phase transformation of the support from γ-Al_2_O_3_ (Figure 1a) to δ-Al_2_O_3_ (Figure 1b) during the HTA. Pd/Al_2_O_3_-Ox and Pd/Al_2_O_3_-Red in Figure 1c,d exhibit characteristic peaks attributed to PdO at 2θ = 29.3°, 33.9°, and 41.9° and 2θ = Pd at 40°, respectively. The crystallite sizes of PdO and Pd estimated using the Scherrer equation are 15 nm and 30 nm, respectively. In contrast, the STEM images and particle size distribution displayed in Figure 2 clearly show significantly large PdO and Pd clusters of size 70 nm for oxidation and 53 nm for reduction. Datye et al. proposed that large metallic Pd clusters form polycrystalline metastable Pd/PdO structures during phase transformation [18,19]. Therefore, it can be interpreted that some of palladium during HTA is auto-reduced [20,21] and forms a polycrystalline Pd/PdO structure, which is consistent with the residual metallic Pd clusters shown in Figure 1c. The size discrepancy between the XRD and TEM results also elucidates that the Pd/PdO structure is polycrystalline with several grain boundaries, as shown in Appendix A. Based on the XRD and TEM results, the significant size discrepancy between PdO and Pd is caused by the polycrystalline nature of the Pd/PdO structure.

### 2.2. Catalytic Behaviors of Pd/Al_2_O_3_ Catalysts

To investigate the difference in effects of oxidative and reductive pretreatments on catalytic behavior, we performed the CH_4_ oxidation reaction at 340 °C for 8 h, as shown in Figure 3. Significantly, the initial activity was comparable regardless of whether it was the oxidative or reductive pretreatment. Accordingly, it can be inferred that the active sites of Pd/Al_2_O_3_-O_x_ and Pd/Al_2_O_3_-Red are most likely to be similar under the reaction conditions. Notably, the decrease in the catalytic activity of Pd/Al_2_O_3_-Red was slower than that of Pd/Al_2_O_3_-O_x_ over 8 h of CH_4_ oxidation. This indicates that the reductive pretreatment of Pd/Al_2_O_3_ catalysts can improve their durability for CH_4_ oxidation compared to oxidative pretreatment. To understand the reason for this observation, we characterized the bulk and surface properties under the reaction conditions through CO titration, in situ XRD, DRIFTS, and STEM. 

We should point out that, in the oxidized sample, the catalytic activity decreases from 40% to around 28% after 8 h of reaction. The issue with the deactivation of the Pd catalyst in the investigation reaction is complex and generally agreed to be related to the presence of water vapor [22,23,24,25,26,27,28,29,30,31], which forms during methane oxidation. Water competes with methane for the active sites on the catalyst surface, since it can dissociate and block the catalytically active undercoordinated Pd sites, forming a spectator Pd-OH species; the effective removal of strongly adsorbed OH species from the PdO surface occurs only at temperatures in excess of 500 °C [31,32].

### 2.3. Comparison of Pd and PdO under the Reaction Conditions

To investigate the transformation of metallic Pd and PdO during the reaction, we used CO titration after the reaction had progressed for a certain period of time to quantify the oxygen to ascertain the relative proportion of PdO in the Pd/PdO cluster, as shown in Figure 4. The reaction conditions for CH_4_ oxidation are the same as those described above, and CO titration was also performed at the same temperature after sufficiently purging with He. The initial analysis of Pd/Al_2_O_3_-O_x_ showed 100% of PdO, and it did not change during the course of the reaction over a period of 8 h. Therefore, there is no change in PdO during the CH_4_ oxidation reaction. However, interestingly, for Pd/Al_2_O_3_-Red, the PdO contents gradually increased as the reaction progressed and plateaued at ~44% of PdO after 3 h of reaction, which is consistent with the previous observations that metallic Pd is oxidized during the reaction [14]. Moreover, during CH_4_ oxidation at 340 °C, small-sized Pd clusters (<3 nm) completely oxidize to PdO(Figure 5). Based on these results, we inferred that metallic Pd is oxidized under the reaction conditions, forming a PdO layer on top of the metallic Pd clusters. The thickness of the PdO phase is ~1.3–1.5 nm. Thus, in the case of smaller crystallites whose size is not sufficient to form the surface PdO layer on metallic Pd clusters, Pd is fully oxidized to PdO. 

To confirm this inference, we performed in situ XRD experiments with different pretreatments under the reaction conditions (Figure 6). Both catalysts exhibit common peaks, which are attributed to δ-Al_2_O_3_. XRD patterns for Pd/Al_2_O_3_-Red demonstrate only metallic Pd at 2θ = 40°, but Pd/Al_2_O_3_-O_x_ shows PdO at 2θ = 33.9° with a little metallic Pd at 2θ = 40°, as mentioned above (Figure 6a). After the reaction, the XRD patterns of Pd/Al_2_O_3_-O_x_ did not change, which indicates that PdO does not change during the reaction, which is in agreement with the CO titration results. In contrast, for Pd/Al_2_O_3_-Red, a broad PdO phase was formed, and simultaneously, a slight decrease in the metallic Pd content was observed during the reaction (Figure 6b). Based on the CO titration results, the broad PdO phase is attributed to the ~44% of PdO generated during the reaction. 

Furthermore, to confirm the generation of the surface PdO layer, we characterized the surface compositions using DRIFTS after CO adsorption, as shown in Figure 7. The catalysts were tested over a period of 8 h of CH_4_ oxidation under the same conditions in DRIFTS cells, and subsequently, IR spectra were collected after CO adsorption at RT following He purging. The IR spectrum of Pd/Al_2_O_3_-O_x_ obtained before the reaction shows a representative band of PdO_x_ at 2140 cm^−1^ [12] and a broad band at 2100–2090 cm^−1^ attributed to metallic Pd, which was partially reduced during CO adsorption [16]. The IR spectrum of Pd/Al_2_O_3_-Red collected before the reaction shows a distinctive peak at 2080 cm^−1^, which is attributed to linear-bound CO, and two peaks at 1980 and 1940 cm^−1^, which are attributed to bridge-bound CO on metallic Pd. The IR spectrum of Pd/Al_2_O_3_-O_x_ obtained after 8 h of reaction (Figure 7a) shows a similar profile to that before the reaction, indicating that the initially formed PdO does not change during the reaction, which is in good agreement with the above results. However, on Pd/Al_2_O_3_-Red, partially oxidized PdO_x_ at 2140 cm^−1^ appeared with the disappearance of bridge-bound CO on metallic Pd, yielding a similar spectrum to that of Pd/Al_2_O_3_-O_x_. Notably, regardless of whether the initial state was metallic Pd or PdO, the catalysts showed the same topmost surface after the reaction. 

In addition, as shown in Appendix A, the surface transformation from Pd to PdO occurred within 3 min of the reaction. Based on these results, we interpreted that the immediately generated topmost PdO_x_ during the reaction is the active site for CH_4_ oxidation, and the similarity in the initial activity of Pd/Al_2_O_3_-O_x_ and Pd/Al_2_O_3_-Red in Figure 3 is due to the presence of the same active site—the topmost PdO_x_ surface. STEM on Pd/Al_2_O_3_-Red after 8 h of reaction also demonstrated the formation of a PdO surface layer, as shown in Figure 8. The high magnification image of Pd/Al_2_O_3_-Red-Rxn 8 h reveals PdO lattice fringes of 0.308 nm and 0.258 nm, corresponding to the PdO(100) and (101) directions, respectively.

Based on the in situ XRD, DRIFTS, and STEM results, Pd/Al_2_O_3_-Red after CH_4_ oxidation reaction consists of a metallic Pd core with a surface PdO layer of 44%. This interpretation is consistent with the findings of Lee et al., who reported that the surface PdO layer on the metallic Pt core enhances the activity and long-term durability of CH_4_ oxidation [33]. Furthermore, it is also related to the report of Mehar et al. that multilayer PdO(101) on Pd(100) exhibited a better catalytic performance than single-layer PdO(101) [17]. Based on these results, we thought that the formation of the PdO layer on the metallic Pd core is the reason for the improved catalytic stability of Pd/Al_2_O_3_-Red shown in Figure 3. In Figure 6b, metallic Pd is dominant after the reaction, and someone can consider this as evidence that metallic Pd might be the active site. However, it is important to note that, regardless of whether it is the PdO or PdO/Pd core–shell structure, both catalysts have practically the same topmost active surface of PdO_x_. 

### 2.4. DFT Modeling of the Interaction of O Species with Pd(111) Surface and Pd_79_ Nanoparticle Models

Periodic DFT calculations were employed to clarify our experimental results. First, we compared the adsorption of one O atom on all available surfaces and subsurface positions of the Pd(111) model (Figure 9). The most stable position was found to be the surface fcc, as the O species located initially at the br and oss positions also moved during the geometry optimization to fcc. The other three-fold hollow position, hcp, is less stable by 0.18 eV than the fcc position. The destabilization of the subsurface positions, tss and tss’, is much higher at 1.32 and 1.95 eV, respectively, with respect to the fcc position. 

To explore the coverage effect, we also considered models with high O surface coverage, where nine O centers are located on the available three-fold (fcc or hcp) surface sites. Similar to the situation with one O center, the model with nine O centers in fcc positions is more stable by 1.74 eV (0.19 per O center) than the model with nine hcp O centers. Further, we considered models with high surface and subsurface O coverage. The most stable configuration was found with nine O_fcc_ and nine O_tss_, as it is more stable by 1.34 eV (0.07 eV per O center) than the structure with nine O centers located at the fcc and tss’ positions. The structure with the initial configuration 9O_hcp_9O_oss_ converts into the most stable one, 9O_fcc_9O_tss_, during the geometry optimization. Thus, on the surface saturated by O Pd(111) and Pd_79_ models considered below, we positioned the surface and subsurface O species at the most stable fcc and tss positions, respectively.

Further, we also modeled the process of O migration in the Pd_79_ nanoparticle model from the center of the (111) facet (fcc-cen position) to the subsurface region (oss-cen position) at low and high O surface coverages, where only one fcc-cen position and all fcc surface positions (62 in total) are occupied by O centers, respectively (Figure 10, Table 1). At low coverage, the process is strongly endothermic by 2.26 eV (Table 1). At the initial state (IS) structure, Pd_79_O-IS, the three Pd-O bond lengths are 199 pm. During the subsurface migration, the O center penetrates through a Pd_3_ triangle located at the center of the (111) facet. Initially, Pd–Pd distances in this triangle are 286–287 pm. In the TS structure, the O center is already located below the Pd surface layer, as the Pd–O distances with the closest Pd surface and subsurface atoms are 201–202 and 232–237 pm, respectively (Table 1). The Pd_3_ triangle is slightly extended, as the corresponding Pd–Pd distances are 308–312 pm. It is shrunk back at the Pd_79_O-FS structure, where the Pd–Pd distance, 288 pm, is similar to the corresponding values in the Pd_79_O-IS structure. On the other hand, the positions of the penetrating O center are close at the Pd_79_O-TS and Pd_79_O-FS structures, as the Pd(subsurface)-O distances in the Pd_79_O-FS structure are by ~10 pm shorter, while the Pd(surface)-O bond lengths are by ~10 longer in comparison to the corresponding distances in the Pd_79_O-TS structure. The “late” nature of the TS is also confirmed by the calculated activation barrier, 2.28 eV, which is similar to the endothermicity of the migration process. 

The occupation of all fcc surface positions by O atoms in the Pd_79_O_62_ system leads to expansion of the Pd_79_ NP, as can be seen from the Pd–Pd distances. At the Pd_79_O_62_-IS structure in the Pd_3_ triangle, they are longer by ~13 pm than those at the Pd_79_O-IS structure. Simultaneously, the Pd–O bond lengths are similar at both structures. At the Pd_79_O_62_-TS structure, the penetrating O center is close to the plane defined by the three Pd atoms of the Pd_3_ triangle. In this structure, the Pd–Pd distances are elongated by 40 pm in comparison to the corresponding values at the Pd_79_O_62_-IS structure, while the Pd–O distances are 2–3 pm shorter in the Pd_79_O_62_-TS structure. When the O center arrives in the subsurface layer, it is essentially equidistantly positioned between the surface and subsurface Pd atoms, while the Pd_3_ triangle notably shrinks as the corresponding Pd–Pd are only 277 pm in the Pd_79_O_62_-FS. The population of all surface positions by O leads to a dramatic decrease of the endothermicity of the O penetration process from 2.26 to 0.31 eV, as the activation barrier is also reduced by almost 50%, from 2.28 to 1.17 eV (Table 1). 

### 2.5. DFT Modeling Adsorption of CO Species on Oxidized Pd(111) and Pd Nanoparticle

In our previous work, we reported CO adsorption on different positions of pristine Pd_79_ palladium nanoparticles. Here, we considered CO adsorption on an oxidized palladium surface and nanoparticles. For CO adsorption on the oxidized palladium surface, we used Pd(111) slab models containing O species (Figure 10) at fcc positions. Our calculations showed that, if all nine fcc positions in the unit cell are occupied by O species, the adsorption of CO is not favorable, as the molecule desorbs from the surface during the geometry optimization. The CO can adsorb on the top if at least two neighboring fcc positions are unoccupied, i.e., on the (O_fcc_)_7_/Pd(111) model, though, even in this model, the BE of CO is as low as −0.17 eV. Much stronger CO adsorption can be achieved when three neighboring fcc positions are free, as the BE values are similar for CO adsorbed on the top and hcp positions, −0.99 and −0.92 eV (Table 2). We also considered CO adsorption on Pd(111) models, where O species occupy not only the surface but also all nine tss subsurface positions. Interestingly, the presence of subsurface O species strengthened the interaction of CO with the Pd(111) surface, as the BE values for the structures Pd111/(O_fcc_)_7_(O_tss_)_9__CO_top, Pd111/(O_fcc_)_6_(O_tss_)_9__CO_hcp, and Pd111/(O_fcc_)_6_(O_tss_)_9__CO_top, −0.26, −1.02, and −1.65 eV are higher as absolute values than the corresponding values for the models that do not contain O species in the subsurface region. These results also highlight that CO prefers on-top adsorption at high O surface and subsurface coverage.

On the oxidized Pd_79_ NP model, CO can be adsorbed on low coordinated sites (corners and edges), even when all 62 surface fcc sites are occupied by O species (Figure 11). The BE is −0.49 eV on the most accessible corner sites, while on the edge sites, the BE value is ~ 0 eV, manifesting an unstable complex. The CO can be adsorbed at the terrace site if three neighboring O centers are removed from one of the (111) facets; at the Pd_79_(O_fcc_)_59_/CO_top-cen structure, the binding energy of an adsorbed CO molecule, −1.48 eV, is similar to the CO adsorption on the same terrace site of the pristine Pd_79_ NP. We also considered CO adsorption on the Pd_79_ NP model with all 62 surface fcc sites and 24 subsurface tss sites occupied by O centers. A comparison of the Pd_79_(O_fcc_)_62_/CO_top-cor and Pd_79_(O_fcc_)_62_(O_tss_)_24_/CO_top-cor structures shows that the addition of subsurface O species slightly decreases the BE value from −0.49 to −0.37 eV. The highest BE value is found for the Pd_79_(O_fcc_)_62_(O_tss_)_24_/CO_top-edg model, −0.63 eV, although the BE value in Pd_79_(O_fcc_)_62_/CO_top-edg is ~0 eV. Surprisingly, the geometries of both carbonyl complexes are the same—the Pd center is in a square planar configuration with respect to the neighboring O centers.

### 2.6. DFT Modeling Adsorption of CO Species on PdO(101) and PdO(100) Slab Models

The results obtained for both PdO(101) slab models are similar (Table 1, Figure 12); thus, only these obtained for the larger unit cell will be commented on in the text below. On the surface of the PdO(101) slab model, there are two types of Pd centers with coordination numbers three and four with respect to the closest O centers. We modeled CO adsorption on both surface positions, as the four-coordinated position is unstable, since the CO molecule migrates to a three-coordinated Pd center during geometry optimization. Indeed, in the most stable structure, PdO(101)/CO_top-3c, CO is adsorbed onto a three-coordinated surface Pd center, as the BE value, −1.49 eV, is similar to the corresponding value in the Pd_79_(O_fcc_)_59_/CO_top-cen structure (Table 2). We also optimized one more structure where CO is in the subsurface region, as the C atom interacts with two surface Pd centers and one subsurface Pd, while the O from the CO molecule interacts with one surface and one subsurface Pd centers. However, this complex is very unstable, as the BE is strongly positive at 3.18 eV.

One complex is found when CO is adsorbed onto the PdO(100) surface model, PdO(100)/CO_top-4c, where the CO interacts with a four-coordinated surface Pd center. The energy of CO in this case is only −0.04 eV.

### 2.7. Assignment of the Experimental C-O Vibrational Frequencies

In our previous study, we modeled CO adsorption at various surface sites of pristine Pd_79_ NP [34]. Our results have shown that, when CO is at the top coordination (top-cen, top-edg, and top-cor), the calculated C-O vibrational frequencies, 2013–2023 cm^−1^ (presented also in Table 3), are systematically shifted to lower frequencies by ~60 cm^−1^ than the experimental bands. Similarly, the calculated C-O vibrational mode for CO adsorption at the top position of the Pd(111) surface slab model, 2036 cm^−1^, is 51 cm^−1^ lower than the experimental band at 2087 cm^−1^.

The calculated C-O vibrational frequencies in the Pd79(O_fcc_)_62_/CO_top-cor and Pd_79_(O_fcc_)_62_/CO_top-edg models, 2090 and 2121 cm^−1^, are 77 and 106 cm^−1^ higher than the corresponding values in the O-free models, 2013 and 2015 cm^−1^ (Table 3). A similar increase of the C-O vibrational frequency from 2023 to 2051 cm^−1^ (Table 2) with the covering of the surface of the nanoparticle is also found for the CO adsorption at terrace top-cen sites, despite the fact that the three closest O centers around the Pd atom, which interacts with the CO molecule, are removed. These DFT results show that the increasing of the Pd oxidation state leads to the increase in the vibrational frequencies due to the increase of the σ-donation and electrostatic interaction, and the decrease of the π-back donation interaction between the palladium and CO ligand [35]. This is also in line with our experimental findings that CO bands of metallic Pd are at 2068 and 2087 cm^−1^, while, for the palladium, the oxide systems are at 2143 and 2160 cm^−1^. Interestingly, when CO is adsorbed on the PdO(101) and Pd(100) surfaces, the calculated C-O vibrational frequencies, 2090 and 2131 cm^−1^, are similar to the corresponding values in the oxidized Pd(111) and Pd_79_ models, where the Pd center to which CO is bound is with coordination numbers 3 and 4, respectively. Hence, our calculated C-O vibrational frequencies for the structures with CN = 3 at 2083–2095 cm^−1^ (Table 3) can rationalize the experimental band at 2143 cm^−1^, while the band at 2160 cm^−1^ can be rationalized by the CO on the edge of the oxidized structures, as well as PdO(100)/CO_top-4c, where the calculated C-O vibrational frequencies are at 2120–2121 and 2131 cm^−1^ (Table 3). In these structures, the coordination number of Pd to which CO is bound is 4. These results are in agreement with the previous study of Martin et al. [36], where an IR band at 2132 cm^−1^ is assigned to CO adsorption on the Pd(101) surface.

## 3. Materials and Methods

### 3.1. Catalyst Preparation

Catalysts were prepared via the incipient wetness impregnation method using an aqueous solution of the Pd precursor (Pd(NH_3_)_4_(NO_3_)_2_) on Al_2_O_3_ (PURALOX TH 100/150, Sasol) and dried at 110 °C for 30 min. The cycles of impregnation–drying were repeated 3 times to obtain a homogenous distribution of Pd. The metal loading of the prepared catalysts was 5.5 wt%. The prepared catalysts were calcined at 500 °C under 20% O_2_ in He (total flow rate = 1000 mL/min) for 2 h and hydrothermally aged at 850 °C for 10 h under 5.7% H_2_O in the air. We chose 10 h of HTA at 850 °C to simulate the engine aging for 200 h in a vehicle. Pd/Al_2_O_3_-O_x_ was prepared via calcination (20% O_2_/He) at 500 °C for 2 h, and Pd/Al_2_O_3_-Red was prepared via reduction (10% H_2_/He) at 500 °C for 1 h.

### 3.2. CH_4_ Oxidation Reaction Tests and CO Titration

CH_4_ oxidation reaction tests were performed in a quartz flow reactor using 50 mg of the catalyst at 340 °C. Prior to the catalytic tests, the catalysts were calcined at 500 °C for 1 h in 20% O_2_/He (flow rate = 60 mL/min). Reductive pretreatment was also conducted at 500 °C for 1 h in 10% H_2_/He (flow rate = 60 mL/min). CH_4_ oxidation activity was measured as a function of reaction time with a mixture of 1% CH_4_ and 10% O_2_ in He balance (total flow rate = 60 mL/min). The effluent gases were passed through a cold trap (dry ice) to remove the water produced during the reaction and then analyzed via gas chromatography (GC, Agilent 7820A) using a Carboxen-1010 column and a thermal conductivity detector. CO titration was also conducted by repeated pulse of 10% CO/He (sample loop = 1 mL) at 340 °C, following He purging. The effluent gases were passed through a methanator with 40% Ni/Al_2_O_3_ and analyzed using GC with flame ionization detector.

### 3.3. Catalyst Characterization

In situ DRIFTS experiments were performed using a Nicolet iS10 FTIR spectrometer equipped with a mercury cadmium telluride detector. The catalysts (13 mg) were loaded into a high-temperature reaction chamber (Harrick Scientific, NewYork, US) equipped with ZnSe windows, which were installed in a Praying Mantis diffuse reflectance accessory. Temperature ramping rate and gas flow rate were 10 °C/min and 60 mL/min, respectively. Loaded catalysts were pretreated using 10% O_2_/He gas at 380 °C for 30 min and cooled to room temperature (RT) by flushing with He for 30 min. Thereafter, the chamber was heated again to 380 °C under 10% H_2_/He, followed by He purging for 30 min and subsequent cooling under He flow. The background spectra were collected immediately after pretreatment. Afterward, CO adsorption using 0.2% CO/He was conducted. Each spectrum was obtained after He purging to remove the weakly bound CO and gas-phase CO. All spectra were collected using a resolution of 4 cm^−1^, averaging 64 scans for every 50 s.

XRD patterns were obtained using a high-resolution X-ray diffractometer (Rigaku, Smartlab, Tokyo, Japan) with Cu K_α_ radiation (λ = 1.54 Å) in the step mode between 2θ values of 10° and 80° with a step size of 0.02°/s. The tube voltage and current were 45 kV and 200 mA, respectively.

High-angle annular dark-field scanning transmission electron microscopy (HAADF-STEM) images were obtained using JEM2100F (JEOL) operated at an accelerating voltage of 200 kV.

### 3.4. Computational Details and Models

Periodic DFT calculations were performed using the plane–wave-based Vienna ab initio simulation package (VASP) [37,38]. We employed the exchange–correlation functional suggested by Perdew and Wang (PW91) [39] with an energy cutoff of 415 eV. For a description of the interaction between atomic cores and electrons, we used the projector augmented wave (PAW) method [40,41]. We sampled the Brillouin zone using the scheme of Monkhorst and Pack [42], as the mesh is 5 × 5 × 1 k points for the Pd(111) slab models, while, for the nanoparticle models, as well as the slab PdO(101) and PdO(100) models, only the Γ point was used (see below). The Methfessel–Paxton technique [43] was invoked to accelerate convergence with a width of 0.1 eV, as the final total energies were extrapolated to zero smearing.

The binding energies of the adsorbed CO molecules E_b_ were calculated as E_b_ = −(E_ad_ + E_sub_ − E_ad/sub_), as E_ad_ is the total energy of the gas phase CO, E_sub_ is the total energy of the Pd substrate (either or not containing pre-ad(ab)sorbed species), and E_ad/sub_ is the total energy of the substrate with all the adsorbates in the optimized geometry. Thus, negative E_b_ values indicate a favorable interaction, i.e., energy release during the adsorption process.

The shape of the employed Pd_79_ nanoparticle (NP) model is a truncated octahedral. This structure exhibits eight (111) facets (each of them consisting of 12 surface Pd atoms) and six very small (100) facets (each of them consisting of four surface Pd atoms). The NP model is placed in a cubic cell with a lattice parameter of 2.0 nm.

We also employed an ideal Pd(111) surface model that was represented as five-layer slab repeated in a supercell geometry. The unit cell is (3 × 3); thus, the surface coverage of 1/9 is achieved when one species is adsorbed into the unit cell. The ad/absorbates were bound to one side of the model and were allowed to relax during geometry optimizations together with the three adjacent (“upper”) layers of the substrate. The two “bottom” layers were kept fixed at the theoretical bulk-terminated geometry, where the Pd–Pd distance was 280 pm.

Other models that we employed were ideal PdO(101) and PdO(100) surfaces. In order to investigate the influence of the size of the unit cell in the former case, we considered two unit cells with a parallelepiped shape different in sizes. The larger one consisted of 144 Pd and 144 O atoms organized in a four-layer slab. The unit cell had dimensions: a = 18.773, b = 18.620, and c = 21.800 Å. The smaller one Pd_60_O_60_ was organized in a five-layer slab with dimensions: a = 12.523, b = 9.288, and c = 24.110 Å. The Pd(100) slab surface model was built from 48 Pd and 48 O atoms, with dimensions: a = 9.288, b = 10.885, and c = 22.836 Å, α = β = γ = 90°.

The available three-fold hollow Pd_3_ adsorption sites for O and CO species on the Pd(111) slab and Pd_79_ nanoparticle models were hcp (hexagonal close packed) or fcc (face-centered cubic) types (Figure 1). The center of these sites was below a Pd atom in the first and second subsurface layers, respectively. The subsurface sites beneath hcp and fcc—tetrahedral and octahedral sites, respectively—were denoted as tss and oss. Other adsorption positions available on the surfaces of both considered models were a top and bridge (denoted as br). They were located above a surface Pd and a Pd–Pd bond, respectively. In order to distinguish the surface and subsurface sites in Pd_79_ NP, we added an additional tag for the following site locations: (cen)—at the center of a (111) facet, (edg)—at the edge between two (111) facets, and (cor)—at the corner between two (111) and one (100) facet (see Figure 1). The subsurface tetrahedral sites just below a surface Pd atom were denoted as tss’.

The activation energy barriers for O penetration from the surface to the subsurface region were derived by scanning the potential energy profile along the line that connects the local minima of the initial (surface) and final (subsurface) positions of the O species. Thus, estimations of the transition states (TSs) were searched in a pointwise fashion along the path connecting the adsorption and absorption configurations, where the height of the carbon atom, defined with respect to the most distant frozen metal layer, was fixed, whereas all other degrees of freedom were allowed to fully relax. The normal mode analysis verified that TSs have one imaginary mode that corresponds to O surface↔subsurface movement.

## 4. Conclusions

In this study, we investigated the intrinsic active sites of the Pd/Al_2_O_3_ catalyst for CH_4_ oxidation, which interconverts between metallic Pd and PdO during the reaction. We used CO titration, in situ XRD, in situ DRIFTS, STEM, and periodic DFT calculations for the analysis. CO titration experiment and in situ XRD clearly showed that approximately 44% of PdO_x_ was formed from metallic Pd during the CH_4_ oxidation reaction. In situ DRIFTS revealed that the topmost PdO_x_ surface is formed on the metallic Pd clusters during the reaction. STEM revealed the formation of the PdO layer on the metallic Pd cluster, and the DFT calculations also demonstrated that, in Pd nanoparticles, the migration of O from the surface to the subsurface region can be facilitated when the surface is O pre-covered, indicating the formation of surface PdO_x_ species. Based on all these results, we concluded that, during the CH_4_ oxidation reaction, regardless of whether the initial state is Pd or PdO, the topmost surface is composed of PdO_x_ species, which is the catalytically relevant active phase. Furthermore, CH_4_ oxidation activity profiles revealed that the surface PdO_x_ layer on the Pd core structure improves the catalytic durability of CH_4_ oxidation, which is in agreement with the findings of previous studies. We believe our study will advance the understanding about the active sites of Pd-based catalysts and provide insights into the development of durable Pd-based catalysts for CH_4_ oxidation.

## Figures and Tables

**Figure 1 molecules-28-01957-f001:**
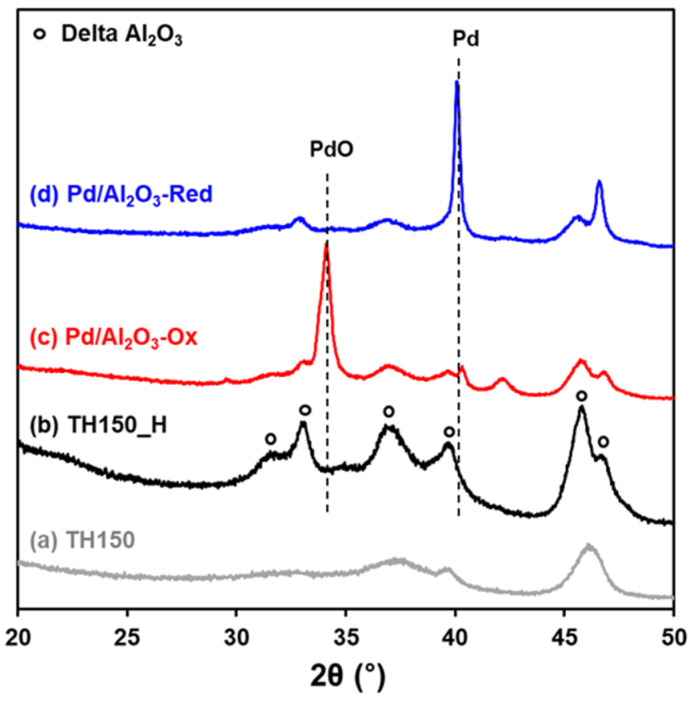
XRD patterns of bare TH150 (**a**) before and (**b**) after HTA, and 5.5 wt% Pd/Al_2_O_3_ with (**c**) oxidative and (**d**) reductive pretreatment.

**Figure 2 molecules-28-01957-f002:**
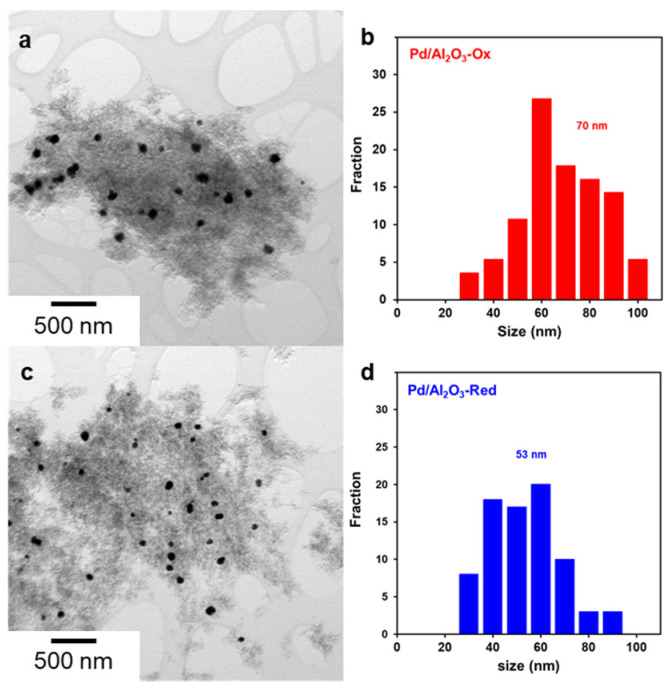
TEM images of (**a**) Pd/Al_2_O_3_-O_x_ and (**c**) Pd/Al_2_O_3_-Red, and particle size distribution of (**b**) Pd/Al_2_O_3_-Ox and (**d**) Pd/Al_2_O_3_-Red.

**Figure 3 molecules-28-01957-f003:**
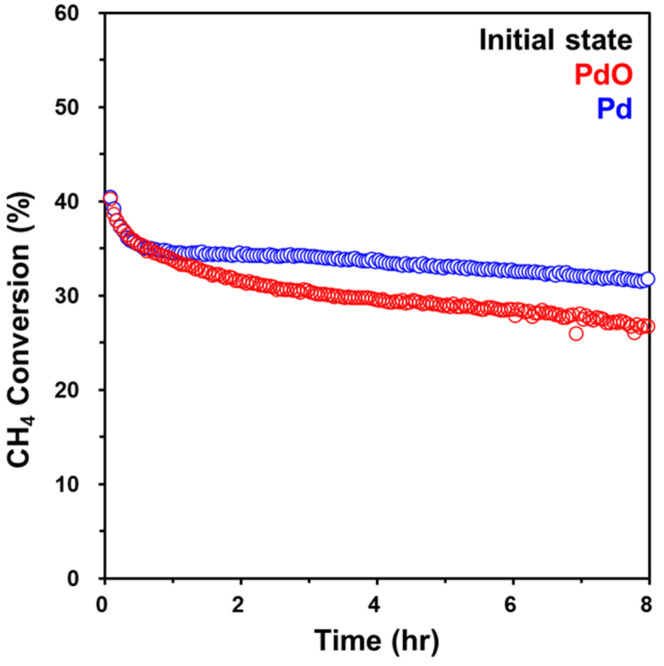
Conversion of CH_4_ oxidation as a function of time for Pd/Al_2_O_3_-O_x_ (red) and Pd/Al_2_O_3_-Red (blue). Reaction conditions: 1% CH_4_; 10% O_2_ in He at 340 °C.

**Figure 4 molecules-28-01957-f004:**
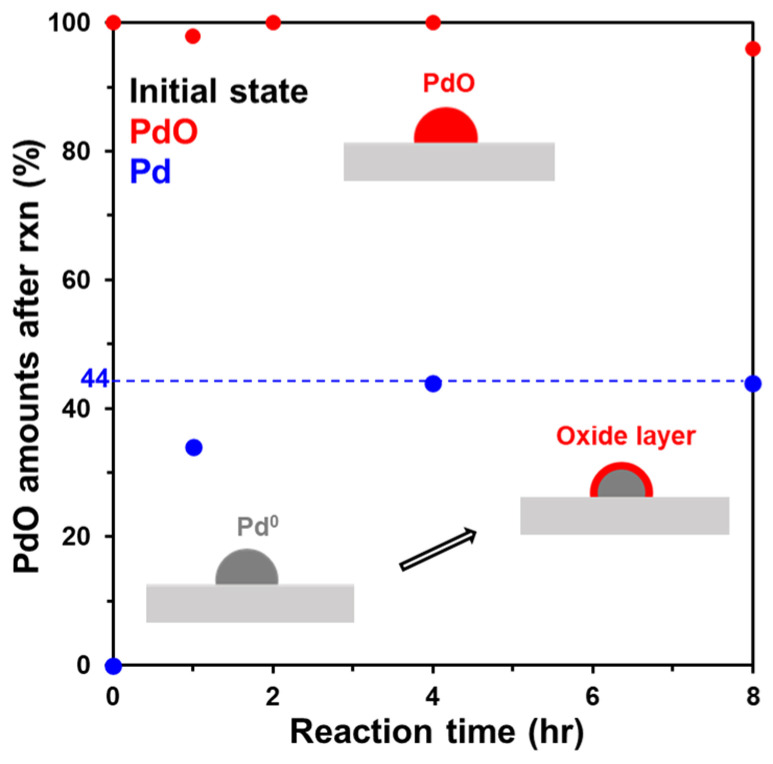
PdO quantity estimated via CO titration as a function of time. Reaction conditions: 1% CH_4_; 10% O_2_ in He at 340 °C; CO titration: 10% CO/He in 1 mL loop at 340 °C.

**Figure 5 molecules-28-01957-f005:**
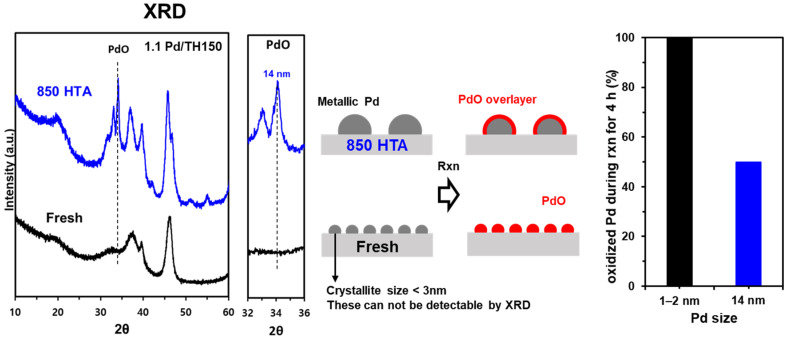
XRD patterns for 1.1 wt% Pd/Al_2_O_3_ before (black) and after HTA (blue) (**left**), scheme for redox behavior of Pd clusters (**middle**), and oxidized Pd during the CH_4_ oxidation of 4 h estimated by CO titration (**right**). As shown in XRD, Pd species were not detected because of their small size (<3 nm) on fresh 1.1 wt% Pd/Al_2_O_3_ catalysts. These small Pd species were fully re-oxidized to PdO during the CH_4_ oxidation of 4 h. Reaction conditions: 1% CH_4_; 10% O_2_ in He at 340 °C.

**Figure 6 molecules-28-01957-f006:**
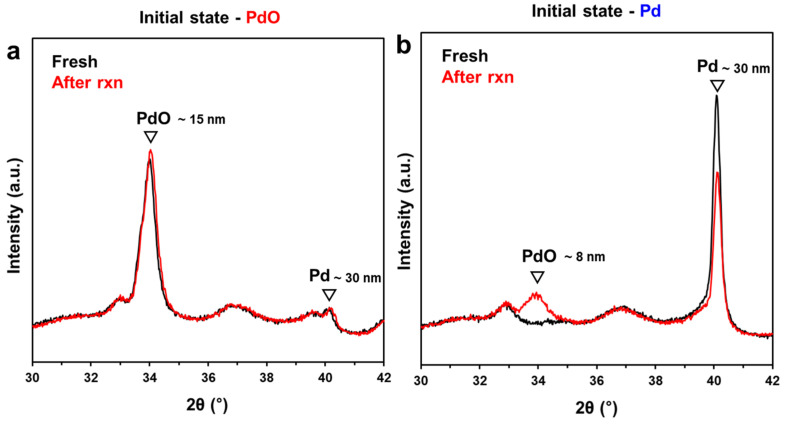
In situ XRD patterns of (**a**) Pd/Al_2_O_3_-O_x_ and (**b**) Pd/Al_2_O_3_-Red before (black) and after (red) 8 h of CO oxidation.

**Figure 7 molecules-28-01957-f007:**
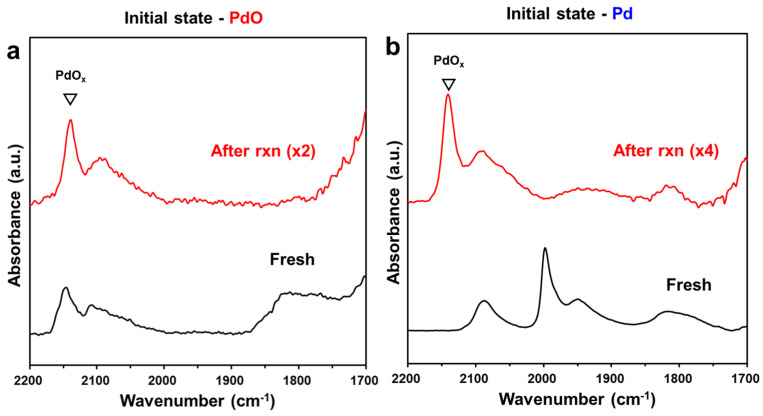
In situ DRIFTS spectra of adsorbed CO on (**a**) Pd/Al_2_O_3_-Ox and (**b**) Pd/Al_2_O_3_-Red before (black) and after (red) 8 h of CO oxidation. DRIFTS spectra were recorded at 25 °C after He purging.

**Figure 8 molecules-28-01957-f008:**
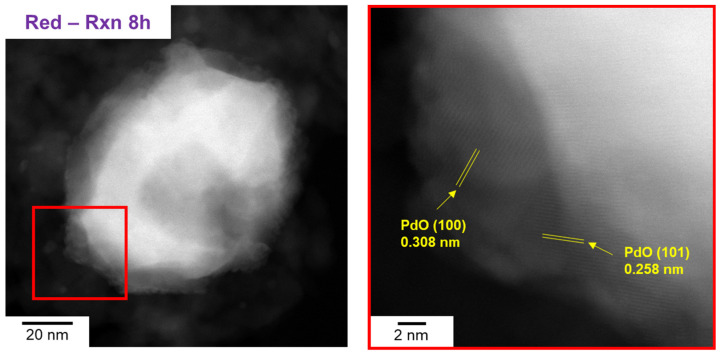
High-magnification STEM images of Pd/Al_2_O_3_-Red-Rxn (8 h).

**Figure 9 molecules-28-01957-f009:**
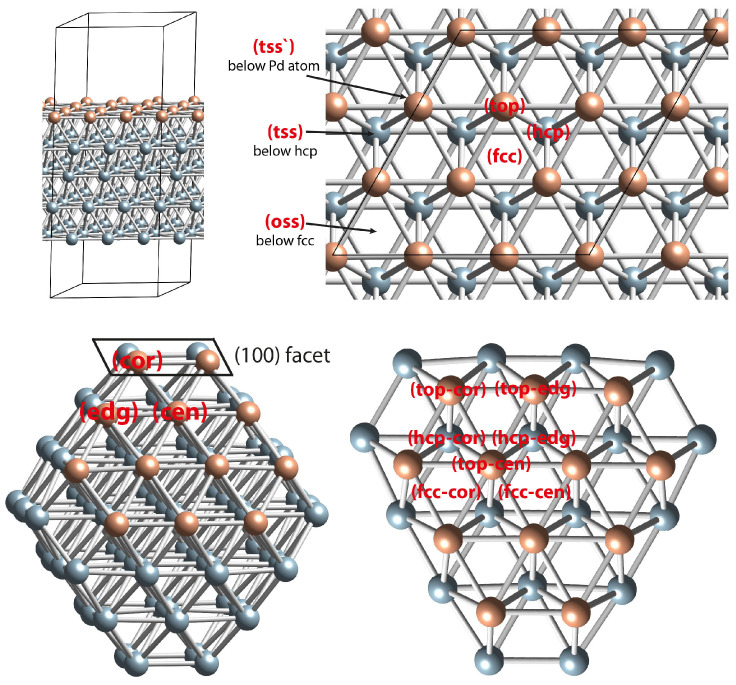
Notation of the site locations for the slab and NP models. Color coding: atoms from Pd top surface layer—orange; other Pd atoms—light blue.

**Figure 10 molecules-28-01957-f010:**
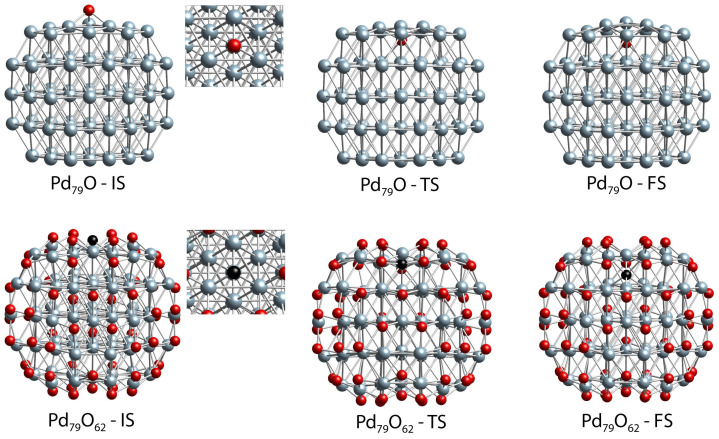
Side and local top (for IS structures only) views of the initial, transition, and final state structures (IS, TS, and FS) for the O_fcc-cen_ subsurface diffusion to the oss-cen position in the Pt_79_O and Pd_79_O_62_ models. Color coding: Pd—blue; O—red; the penetrating O_fcc-cen_ atom in Pd_79_O_62_ models—black.

**Figure 11 molecules-28-01957-f011:**
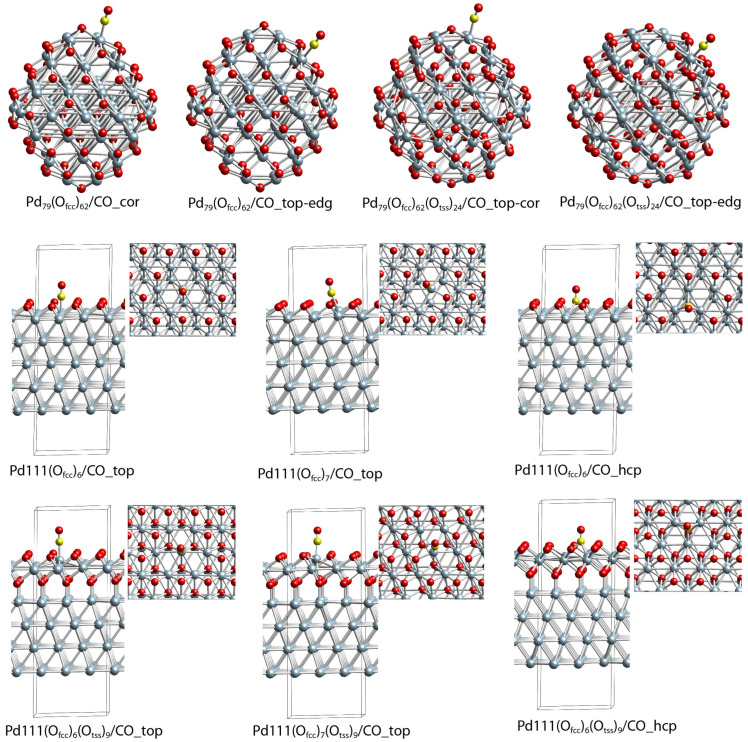
Adsorption complexes of CO on the O-containing Pd(111) slab and Pd_79_ nanoparticle models. Color coding: light blue—Pd, red—O, and yellow—C. In the case of the Pd(111) surface also, the top view is shown as only the first two Pd layers are presented.

**Figure 12 molecules-28-01957-f012:**
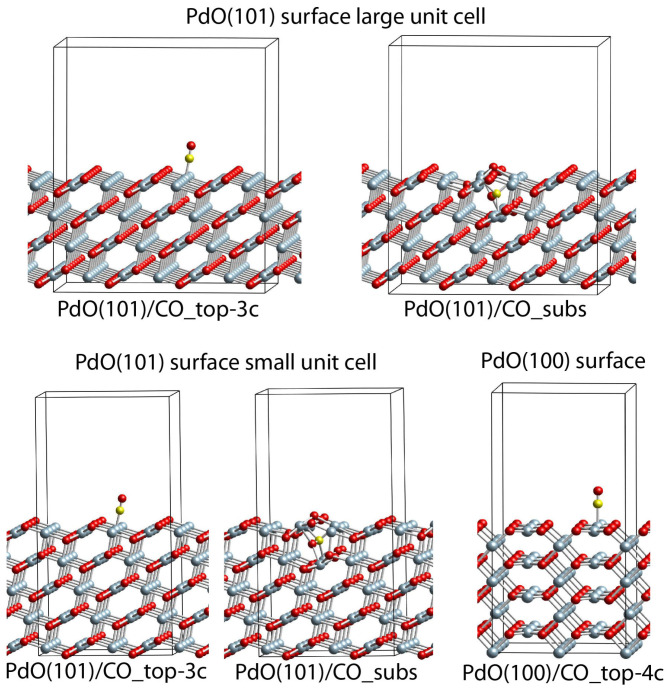
Adsorption complexes of CO on the PdO(101) and PdO(100) slab models. Color coding: light blue—Pd, red—O, and yellow—C.

**Table 1 molecules-28-01957-t001:** Energy and activation barriers (in eV) of the O migration from the surface to the subsurface region at the center of the (111) facet of Pd_79_ NP at low and high O surface coverage. Structural characteristics (Pd–O and Pd–Pd distances in pm) of the initial, transition, and final state structures are also shown.

Structure	E^≠^	ΔE	Pd-O	Pd-Pd
Pd_79_O-IS	2.28	2.26	199 *^a^*;199 *^a^*;199 *^a^*	286;287;287
Pd_79_O-TS			201 *^a^*;202 *^a^*;202 *^a^*/232 *^b^*;233 *^b^*;237 *^b^*	308;310;312
Pd_79_O-FS			210 *^a^*;210 *^a^*;211 *^a^*/222 *^b^*;223 *^b^*;223 *^b^*	288;288;288
Pd_79_O_62_-IS	1.17	0.31	198 *^a^*;198 *^a^*;198 *^a^*	298;298;298
Pd_79_O_62_-TS			195 *^a^*;196 *^a^*;196 *^a^*	338;339;340
Pd_79_O_62_-FS			218 *^a^*;218 *^a^*;219 *^a^*/216 *^b^*;216 *^b^*;216 *^b^*	277;277;277

*^a^* Pd–O distances that include surface Pd atoms; *^b^* Pd–O distances that include subsurface Pd atoms.

**Table 2 molecules-28-01957-t002:** Calculated energetic and structural characteristics of the adsorption complexes of one CO molecule on the oxidized Pd(111) slab model and palladium nanoparticles, as well as PdO(101) and Pd(100) slab models. Energy values are in eV, distances in pm, and vibrational frequencies in cm^−1^.

Structure	BE *^a^*	CN *^b^*	r(Pd-C)	ν(C-O)	Δr(C-O) *^c^*
**Oxidized palladium surface**					
Pd111(O_fcc_)_7_/CO_top	−0.17	9	194	2085	0.5
Pd111(O_fcc_)_6_/CO_hcp	−0.92	9;9;9	211;211;211	1873	3.4
Pd111(O_fcc_)_6_/CO_top	−0.99	9	192	2072	0.8
Pd111(O_fcc_)_7_(O_tss_)_9_/CO_top	−0.26	9	189	2090	0.7
Pd111(O_fcc_)_6_(O_tss_)_9_/CO_hcp	−1.02	9;9;9	214;214;215	1889	3.2
Pd111(O_fcc_)_6_(O_tss_)_9_/CO_top	−1.65	9	187	2083	0.9
**Oxidized palladium nanoparticle**					
Pd_79_(O_fcc_)_62_/CO_top-cor	−0.49	6	200	2088	0.4
Pd_79_(O_fcc_)_62_/CO_top-edg	0.04	7	203	2121	−0.1
Pd_79_(O_fcc_)_59_/CO_top-cen	−1.48	9	188	2051	1.2
Pd_79_(O_fcc_)_62_(O_tss_)_24_/CO_top-cor	−0.37	6	201	2095	0.2
Pd_79_(O_fcc_)_62_(O_tss_)_24_/CO_top-edg	−0.63	7	194	2120	0.0
**PdO(101) surface large unit cell**					
PdO(101)/CO_top-3c	−1.49	3	189	2090	0.7
PdO(101)/CO_subs *^d^*	3.18	2;4;4	203;214;218	1540	7.2
**PdO(101) surface small unit cell**					
PdO(101)/CO_top-3c	−1.56	3	189	2084	0.8
PdO(101)/CO_subs *^e^*	3.13	2;4;4	201;214;218	1499	8.1
**PdO(100) surface**					
PdO(100)/CO_top-4c	−0.04	4	203	2131	−0.1

*^a^* BE per CO adsorbate; *^b^* Coordination number of the Pd atom, on which the CO molecule is adsorbed with respect to the neighboring Pd atoms for the Pd_79_ and Pd(111) models and with respect to the neighboring O atoms for the PdO(101) and PdO(100) models; *^c^* Elongation of the C-O bond upon adsorption with respect to the calculated bond length in the gas-phase CO of 114.2 pm; *^d^* O atom from the CO molecule interacts with two four-coordinated (surface and subsurface) Pd atoms, as the Pd–O distances are 224 and 229 pm, respectively; *^e^* O atom from the CO molecule interacts with two four-coordinated (surface and subsurface) Pd atoms, as the Pd–O distances are 221 and 228 pm, respectively.

**Table 3 molecules-28-01957-t003:** Assignment of the C-O vibrational frequencies (in cm^−1^) for CO adsorption on the Pd and PdO_x_ models and the experimental bands. The differences between the experimental and theoretical values (ν_exp_–ν_calc_) are given in parentheses.

Structures	CN *^a^*	ν_calc_	ν_exp_
*On-top slab:* Pd111/CO_top		2036	2087 (51)
*On-top terrace:* Pd_79_/CO_top-cen *^b^*		2023	2087 (64)
*On-top edge:* Pd_79_/CO_top-edg *^b^*		2015	2068 (53)
*On-top corner:* Pd_79_/CO_top-cor *^b^*		2013	2068 (55)
Pd111(O_fcc_)_6_(O_tss_)_9_/CO_top	3	2083	2143 (60)
Pd111(O_fcc_)_7_/CO_top	3	2085	2143 (58)
Pd111(O_fcc_)_7_(O_tss_)_9_/CO_top	3	2090	2143 (53)
Pd_79_(O_fcc_)_62_/CO_top-cor	3	2088	2143 (55)
PdO(101)/CO_top-3c	3	2090	2143 (53)
Pd_79_(O_fcc_)_62_(O_tss_)_24_/CO_top-cor	3	2095	2143 (48)
Pd_79_(O_fcc_)_62_/CO_top-edg	4	2121	2160 (39)
Pd_79_(O_fcc_)_62_(O_tss_)_24_/CO_top-edg	4	2120	2160 (40)
PdO(100)/CO_top-4c	4	2131	2160 (29)

*^a^* Coordination number of the Pd atom, on which the CO molecule is adsorbed with respect to the neighboring O atoms; *^b^*—data from Ref [29].

## Data Availability

Data are available by the authors.

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
