# Peer review of "Understanding of Active Sites and Interconversion of Pd and PdO during CH4 Oxidation"

_molecules, 2023, doi:10.3390/molecules28041957_

Round 1

Reviewer 1 Report (Previous Reviewer 1)

I thank the authors for his/her reply. The role of water in deactivation is interesting and could explain the observed catalytic trend. The manuscirpt has improved significantly covering important aspects not only on activity but also on long term catalyst stability.

The topic is of interest. In many cases, active sites are generated in situ under reaction conditions, which has been scarcely reported up to now.

The authors provide direct evidence of an In situ formation of active sites。 The methodology is correct. In the present work the authors studied the active sites on Pd based catalysts in CH4 oxidation. They found an in situ oxidation of Pd metal to Pd oxide, behaving the last one as active sites.

With this new input, I consider the manuscript suitable for publication.

Author Response

I thank the authors for his/her reply. The role of water in deactivation is interesting and could explain the observed catalytic trend. The manuscirpt has improved significantly covering important aspects not only on activity but also on long term catalyst stability.

The topic is of interest. In many cases, active sites are generated in situ under reaction conditions, which has been scarcely reported up to now.
The authors provide direct evidence of an In situ formation of active sites。 The methodology is correct. In the present work the authors studied the active sites on Pd based catalysts in CH4 oxidation. They found an in situ oxidation of Pd metal to Pd oxide, behaving the last one as active sites.

With this new input, I consider the manuscript suitable for publication.

Response: We thank the reviewer for the valuable comments and positive evaluation of our manuscript.

Reviewer 2 Report (New Reviewer)

Unfortunately I cannot find in this manuscript a clear aim of this work. Process of PdO formation on Pd during the oxidation reaction and theoretical calculation of active centres were known a long time ago. Also in this work I did not find  any new principal information about this system. For this reason I cannot recommend this manuscript for publication.

Author Response

Unfortunately I cannot find in this manuscript a clear aim of this work. Process of PdO formation on Pd during the oxidation reaction and theoretical calculation of active centres were known a long time ago. Also in this work I did not find  any new principal information about this system. For this reason I cannot recommend this manuscript for publication.

Response: We thank the reviewer for his/her time to review our manuscript. However, we disagree with his/her opinion. Pd to PdO conversion during methane oxidation is known in the literature but the process itself and the level of oxidation is not well understood, yet. In our manuscript, it was found that at the proposed experimental conditions the level of Pd oxidation is significant (equilibrated at ~44%). To the best of our knowledge this is the first study where it was found that during the CH4 oxidation reaction, regardless of whether the initial state is Pd or PdO, the topmost surface is composed of PdOx species, which is the catalytically relevant active phase. The calculated Pd oxide layer thickness implies subsurface diffusion of oxygen and this provides new insight for understanding of the dynamic behavior of Pd based methane oxidation catalysts. Thus, we believe our study will advance the understanding of the active sites of Pd-based catalysts and provide insights into the development of durable Pd-based catalysts for CH4 oxidation.

Reviewer 3 Report (Previous Reviewer 2)

Dong, Hristiyan and Co-Workers described a «Understanding of active sites and interconversion of Pd and PdO during CH4 oxidation». The article is well written, all methods are relevant and necessary to describe the results and conclusions obtained in this work.

 While reading, there were a few questions and comments:

1) Remove yellow text highlight: line 115-122 and links 22 to 33.

2) In Figure 11, it is desirable to increase the text (different font sizes).

3) Figure S2 from the supplementary would be better added to the article. In my opinion, it is quite informative and I think it is necessary in the article.

4) What is the thickness of the oxide layer in PdO?

Author Response

Dong, Hristiyan and Co-Workers described a «Understanding of active sites and interconversion of Pd and PdO during CH4 oxidation». The article is well written, all methods are relevant and necessary to describe the results and conclusions obtained in this work.

Response: We thank the reviewer for the valuable comments and positive evaluation of our manuscript.

 While reading, there were a few questions and comments:

1) Remove yellow text highlight: line 115-122 and links 22 to 33.

Response: The suggested change is done.

2) In Figure 11, it is desirable to increase the text (different font sizes).

Response: As suggested we increased the font size of the text in Figure 11.

3) Figure S2 from the supplementary would be better added to the article. In my opinion, it is quite informative and I think it is necessary in the article.

Response: Following the suggestion, we moved Figure S2 from SI to the manuscript (now it is Figure 5, we also changed the numeration of Figures 5-11).

4) What is the thickness of the oxide layer in PdO?

Response: The thickness of PdO phase is ~1.3-1.5 nm under the experimental conditions of this manuscript. To clarify this issue in the text we changed the following sentences at the end of p. 4: “Based on these results, we inferred that the metallic Pd is oxidized under the reaction conditions, forming a PdO layer on top of the metallic Pd clusters. Moreover, in the case of smaller crystallites whose size is not sufficient to form the surface PdO layer on metallic Pd clusters, Pd is fully oxidized to PdO.” to “Based on these results, we inferred that the metallic Pd is oxidized under the reaction conditions, forming a PdO layer on top of the metallic Pd clusters. The thickness of the PdO phase is ~1.3-1.5 nm. Thus, in the case of smaller crystallites whose size is not sufficient to form the surface PdO layer on metallic Pd clusters, Pd is fully oxidized to PdO.”

This manuscript is a resubmission of an earlier submission. The following is a list of the peer review reports and author responses from that submission.

Round 1

Reviewer 1 Report

In the present work, the role of oxidised Pd species as active sites for CH4 oxidation has been elucidated combining catalytic with spectroscopic studies. The nature of active sites are widely discussed in the literature being the author’s objective to shed light on the ambiguity of the literature.  In this direction, the work is clear, concise and well written. According to the authors, Pd metal nanoparticles are easily oxidised under reaction conditions, even after 3min. This explain the same initial catalytic activity, whether the initial state of the catalyst was metallic Pd or PdO. However, one point which remain unclear is why the activity decrease (Figure 3). Thus, in the oxidised sample (i.e., starting from PdO), the catalytic activity decrease from 40% to around 28% after 8 h of reaction. However no apparent change have been observed from XRD and IR-CO. Moreover, it is also unclear, why in the reduced sample, if the PdO content increase gradually under reaction conditions, starting from Pd metal and reaching a plateaued of 44% PdO after 3h of reaction (line 126-128), the activity decrease from 40% to 35%.  These points rise some questions about the assignation of oxidised Pd species as active sites. Probably the role of surface, subsurface oxygen is critical and need to be discussed in more detail.

Another question is the metal loading of the catalyst and what is the color code in figure 8.

In conclusion, the manuscript need to be revised before publication.

Author Response

In the present work, the role of oxidised Pd species as active sites for CH4 oxidation has been elucidated combining catalytic with spectroscopic studies. The nature of active sites are widely discussed in the literature being the author’s objective to shed light on the ambiguity of the literature.  In this direction, the work is clear, concise and well written. According to the authors, Pd metal nanoparticles are easily oxidised under reaction conditions, even after 3min. This explain the same initial catalytic activity, whether the initial state of the catalyst was metallic Pd or PdO.

Answer: We thank the reviewer for the valuable comments and positive evaluation of our manuscript.

However, one point which remain unclear is why the activity decrease (Figure 3). Thus, in the oxidised sample (i.e., starting from PdO), the catalytic activity decrease from 40% to around 28% after 8 h of reaction. However no apparent change have been observed from XRD and IR-CO. Moreover, it is also unclear, why in the reduced sample, if the PdO content increase gradually under reaction conditions, starting from Pd metal and reaching a plateaued of 44% PdO after 3h of reaction (line 126-128), the activity decrease from 40% to 35%.  These points rise some questions about the assignation of oxidised Pd species as active sites. Probably the role of surface, subsurface oxygen is critical and need to be discussed in more detail.

Answer: The reviewer raised one very important yet unsolved question/problem for the catalytic community concerning the practical application of methane oxidation catalysts. Indeed, these methane oxidation catalysts always required a continuous repeated regeneration process for practical application due to such deactivation. Although the topic is debated intensively there is no clear explanation for why these catalysts are deactivated persistently.

One possible direction for searching for a solution to the problem of why the reduced sample is more durable than the oxidized sample might be related to the previous report on PtPd bimetallic catalysts. According to а previous work (Lee et al. Appl. Catal. B: Env. 260 (2020) 118098), metallic Pt core with PdO surface structure is more durable than homogeneous PtPd alloy. Consistently in our work, we inferred that metallic Pd core with PdO surface structure is more durable than PdO system. This interpretation is mentioned in line 181/190 “Based on the in-situ XRD, DRIFTS, and STEM results, Pd/Al2O3-Red after CH4 oxidation reaction consists of a metallic Pd core with a surface PdO layer of 44%. This interpretation is consistent with the findings of Lee et al., who reported that the surface PdO layer on the metallic Pt core enhances the activity and long-term durability for CH4 oxidation [22]. Furthermore, it is also related with the report of Mehar et al. that multilayer PdO(101) on Pd(100) exhibited the better catalytic performance than single-layer PdO(101) [17]. Based on these results, we thought that the formation of the PdO layer on the metallic Pd core is the reason for the improved catalytic stability of Pd/Al2O3-Red shown in Figure 3.”

Another question is the metal loading of the catalyst

Answer: The information is provided in line 367/368: “The metal loading of the prepared catalysts was 5.5 wt%.”

and what is the color code in figure 8.

In conclusion, the manuscript need to be revised before publication.

Answer: Color coding: atoms from Pd top surface layer – orange; other Pd atoms – light blue. The clarification is added in the figure caption (line 203/204).

Reviewer 2 Report

Dong, Hristiyan and Co-Workers described a «Understanding of active sites and interconversion of Pd and PdO during CH4 oxidation». The article is well written, all methods are relevant and necessary to describe the results and conclusions obtained in this work.

While reading, there were a few questions and comments:

1) The abbreviation HTA must be deciphered where it is mentioned for the first time (line 79, not 367)

2) In paragraph 3.1 Catalyst preparation, it is necessary to clearly describe how the catalyst containing PdO and Pd was obtained.

3) In figure 1c, explain the reflexes in the area 2θ ≈ 29° and 42° , what do they refer to?

Author Response

Dong, Hristiyan and Co-Workers described a «Understanding of active sites and interconversion of Pd and PdO during CH4 oxidation». The article is well written, all methods are relevant and necessary to describe the results and conclusions obtained in this work.

Answer: We thank the reviewer for the valuable comments and positive evaluation of our manuscript.

While reading, there were a few questions and comments:

1) The abbreviation HTA must be deciphered where it is mentioned for the first time (line 79, not 367)
Answer: We apologize for our mistake. We reflected the suggestion in the revised manuscript.

2) In paragraph 3.1 Catalyst preparation, it is necessary to clearly describe how the catalyst containing PdO and Pd was obtained.

Answer: Following the Reviewer's suggestion, we added in the revised version a description of catalyst preparations as below.

Line 368 "Pd/Al2O3-Ox was prepared via calcination (20% O2/He) at 500 °C for 2 h and Pd/Al2O3-Red was prepared via reduction (10% H2/He) at 500 °C for 1 h."

3) In figure 1c, explain the reflexes in the area 2θ ≈ 29° and 42°, what do they refer to?

Answer: We apologize for the insufficient explanation. The reflexes at 2θ ≈ 29° and 42° come from PdO (ASTM 06-0515) as shown in the figure below. The (100) and (110) planes of PdO would diffract at 2θ ≈ 29° and 42°, respectively. Additional explanation has been added to the revised manuscript as follows.

Line 84 "Pd/Al2O3-Ox and Pd/Al2O3-Red in Figure 1(c) and (d) exhibit characteristic peaks attributed to PdO at 2θ = 29.3°, 33.9° and 41.9° and 2θ = Pd at 40°, respectively.”

Round 2

Reviewer 1 Report

Thanks for the authors reply, but the main question risen by the reviewer, regarding the desactivation of the reduced sample, is not answered. According to spectroscopic studies, the PdO content increase gradually under reaction conditions, reaching a plateau after 3h of reaction resulting in a 44 % PdO. Taken into account the arguments given by the authors that should result in an increase in the catalytic activity, which is not the case.  In fact, Mehar et al, report that multilayer PdO (101) on Pd(100) exhibit better catalytic performance than single-layer PdO (101).

This contradictory behaviour indicate that something else is contributing to the catalytic activity which remains yet unresolved. Since the topic of the paper is related to surface oxidation behaving these as real active sites, I consider that the paper as it is, cannot be published. The authors need to do an effort in order to disentangle additional features influencing the catalytic performance of the samples. 

Sorry  not to be positive at all, but the topic is interesting, but need to be accurately investigated.

Author Response

We thank the reviewer for his/her comments, however, we were disappointed by his/her decision based on an issue that is far beyond the scope of this manuscript. Please note that this is a scientific manuscript with a focus on the fundamental understanding of active sites and interconversion of Pd and PdO during CH4 oxidation. The industrial application and catalyst deactivation are side issue for this study.

Nevertheless, we tried to explain that the deactivation processes are quite complex for this reaction and yet not clarified in the literature. We investigated the literature once again and have found that the deactivation during methane oxidation is related to water accumulation and dissociation on PdO surfaces. The following text was added to the manuscript (p. 4-5): “We would like to point out that in the oxidized sample, the catalytic activity decreases from 40% to around 28% after 8 h of reaction. The issue with deactivation of Pd catalyst in the investigation reaction is complex and generally agreed to be related to the presence of water vapor [[i],[ii],[iii],[iv],[v],[vi],[vii],[viii],[ix],[x],[xi]] which forms during methane oxidation. Water competes with methane for the active sites on the catalyst surface, since it can dissociate and block the catalytically active undercoordinated Pd sites forming a spectator Pd-OH species; effective removal of strongly adsorbed OH species from PdO surface occurs only at temperatures in excess of 500 °C [11,[xii]].

[i].        Mouaddib, N.; Feumijantou, C.; Garbowski, E.; Primet, M. Catalytic oxidation of methane over palladium supported on alumina: Influence of the oxygen-to-methane ratio. Appl Catal a-Gen 1992, 87 (1), 129-144. https://doi.org/10.1016/0926-860X(92)80177-E

[ii].       Gelin, P.; Primet, M. Complete oxidation of methane at low temperature over noble metal based catalysts: a review.. Appl Catal B-Environ 2002, 39 (1), 1-37. https://doi.org/10.1016/S0926-3373(02)00076-0

[iii].      Eguchi, K.; Arai, H. Recent advances in high temperature catalytic combustion. Catal Today 1996, 29 (1-4), 379-386. https://doi.org/10.1016/0920-5861(95)00308-8

[iv].      Choudhary, T. V.; Banerjee, S.; Choudhary, V. R. Catalysts for combustion of methane and lower alkanes.  Appl Catal a-Gen 2002, 234 (1-2), 1-23. https://doi.org/10.1016/S0926-860X(02)00231-4

[v].       Cullis, C. F.; Willatt, B. M. Oxidation of methane over supported precious metal catalysts.  J Catal 1983, 83 (2), 267-285. https://doi.org/10.1016/0021-9517(83)90054-4

[vi].      Lyubovsky, M.; Pfefferle, L. Complete methane oxidation over Pd catalyst supported on α-alumina. Influence of temperature and oxygen pressure on the catalyst activity. Catal Today 1999, 47 (1-4), 29-44. https://doi.org/10.1016/S0920-5861(98)00281-8

[vii].      Eguchi, K.; Arai, H. Low temperature oxidation of methane over Pd-based catalysts—effect of support oxide on the combustion activity. Appl Catal a-Gen 2001, 222 (1-2), 359-367. https://doi.org/10.1016/S0926-860X(01)00843-2

[viii].     Ciuparu, D.; Lyubovsky, M. R.; Altman, E.; Pfefferle, L. D.; Datye, A. Catalytic combustion of methane over palladium-based catalysts. Catal Rev 2002, 44 (4), 593-649. https://doi.org/10.1081/CR-120015482

[ix].      Lampert, J. K.; Kazi, M. S.; Farrauto, R. J. Palladium catalyst performance for methane emissions abatement from lean burn natural gas vehicles. Appl Catal B-Environ 1997, 14 (3-4), 211-223. https://doi.org/10.1016/S0926-3373(97)00024-6

[x].       Sekizawa, K.; Widjaja, H.; Maeda, S.; Ozawa, Y.; Eguchi, K. Low temperature oxidation of methane over Pd catalyst supported on metal oxides. Catal Today 2000, 59 (1-2), 69-74. https://doi.org/10.1016/S0920-5861(00)00273-X

[xi].      Kovarik, L.; Jaegers. N.; Szanyi, J.; Derewinski, M.; Wang, Y.; Khivantsev, K. PdO self-assembly on zeolite SSZ-13 with rows of O3Al(IV)OH selectively incorporated in PdO(101) facets for moisture-resistant methane oxidation. ChemRxiv. Cambridge: Cambridge Open Engage; 2021; 10.26434/chemrxiv-2021-qrd13

[xii].      Kan, H. H.; Colmyer, R. J.; Asthagiri, A.; Weaver, J. F. Adsorption of Water on a PdO(101) Thin Film: Evidence of an Adsorbed HO−H2O Complex, The Journal of Physical Chemistry C 2009, 113 (4), 1495-1506.